# Obstructive sleep apnea severity varies by season and environmental influences such as ambient temperature

Bastien Lechat [1] ✉, Duc Phuc Nguyen[1], Kelly Sansom[1,2], Lucia Pinilla [1], Hannah Scott [1], Amy C. Reynolds[1], Andrew Vakulin[1], Jack Manners[1], Robert J. Adams[1], Jean-Louis Pepin [3], Pierre Escourrou [4], Peter Catcheside [1] & Danny J. Eckert [1]

## Abstract

**Background** Obstructive sleep apnea (OSA) severity often varies considerably from night-to-night, but whether environmental factors play a role is unclear. This study investigated seasonal and temperature-related changes in OSA severity.

**Methods** Data were acquired from 70,052 participants with an average apnea-hypopnea index (AHI) ≥ 5 events/hour who used an under-mattress sleep sensor at least 4 times/week between January 2020 and September 2023. Fixed effect models were used to investigate the association between AHI and day of the year, adjusting for geographical location, variation in total sleep time, ambient temperature, and air pollution.

**Results** Participants are middle-aged (mean ± SD, 53 ± 13 years), predominantly male (81%), overweight (BMI; 29 ± 6 kg/m²) and have an average of 492 ± 341 nights of data. Mean AHI is 18.0 ± 14.0 events/h and within-subject coefficient of variation is ±51%. AHI is ~5% higher during summer/winter compared to spring/autumn in the northern hemisphere, and 10–15% higher during summer compared to spring in the southern hemisphere. Higher ambient temperature (25th vs. 75th percentiles; 6 vs. 18 degrees Celsius) is associated with a 6.4% (95% CI; 6.3–6.5) increase in AHI. Results are consistent across 23 countries, although the effect of temperature on AHI is larger in Europe vs. the United States or Australia.

**Conclusions** Here we demonstrate a seasonal component to OSA severity, partially explained by ambient temperature and seasonal variation in sleep duration. Our findings highlight the need to report data collection months in OSA clinical trials, and further study to uncover the physiology behind seasonal variation in OSA severity are required.

## Plain language summary

Obstructive sleep apnea (OSA) severity often varies considerably from night-to-night, yet whether there is a seasonal component to OSA severity is unclear. This study used data from 70,052 people who tracked their sleep and OSA severity for up to 3.5 years using an FDA-cleared under-the-mattress sensor. We examined whether the severity of OSA changes with the seasons. We found that OSA severity varies across the year and is partly explained by changes in sleep duration and environmental factors like ambient temperature. These findings highlight the importance of considering the time of year when conducting OSA studies or evaluating treatments. Future research is needed to better understand the physiological mechanisms behind these seasonal changes in OSA severity.

Obstructive sleep apnea (OSA) is the most prevalent sleep-related breathing disorder, estimated to affect 1 billion adults globally[1,2]. OSA severity is typically quantified using the apnea-hypopnoea index (AHI), a count of the number of partial (hypopnoea) or total (apnea) upper airway collapses per hour of sleep. Untreated OSA is associated with a range of adverse health outcomes, road safety events, reduced quality of life, and all-cause mortality[3,4].

There is considerable night-to-night variability in OSA severity which leads to diagnostic misclassification in 20–50% of patients[2,5–7].

High night-to-night variability is an important predictor of poor health outcomes[2,5,8,9] and therefore understanding the factors that contribute to this variability is important. Existing studies suggest a potential seasonal component to OSA severity, although results are conflicting. A cross-sectional analysis of 7000 polysomnographic studies of people suspected of a sleep disorder suggested that the AHI is highest during winter[10]. However, a longitudinal cohort study of 100 participants over ~1 year found that the respiratory disturbance index (derived from pacemakers) is 10% higher during summer versus winter[11].

[1]Flinders Health and Medical Research Institute:Sleep Health, College of Medicine and Public Health, Flinders University, Adelaide, Australia. [2]Centre for Healthy Ageing, Health Futures Institute, Murdoch University, Perth, Western Australia. [3]Univ. Grenoble Alpes, HP2 Laboratory, Inserm U-1300, CHU Grenoble Alpes, Grenoble, France. [4]Centre Interdisciplinaire du Sommeil, Paris, France. ✉e-mail: bastien.lechat@flinders.edu.au

Some studies have focused on environmental factors that may partially explain potential seasonal variation in OSA severity, such as ambient temperature and air pollution[12–15]. However, these have also produced conflicting results which may be explained by differences in the measurement of sleep and environmental variables. A major limitation is that most prior studies are cross-sectional and use single-night estimates of AHI and therefore neglects to account for the substantial night-to-night variability in AHI[2,5,6]. Only one large study used multi-night AHI measurements, but with an unvalidated device[12]. Furthermore, sleep duration and timing varies by season[16,17], which could partly explain seasonal variation in AHI through sleep regularity[18] or circadian effects[19].

In this study, we use multi-night, longitudinal measurements of OSA severity using a validated under-mattress sensor in ~70,000 people with OSA, collected over ~3.5 years across multiple geographical locations to estimate seasonal variations in OSA severity, adjusting for key confounders including environmental factors, sleep duration and timing variability. We show that AHI is ~5% higher during summer/winter compared to spring/ autumn in the northern hemisphere, and 10–15% higher during summer compared to spring in the southern hemisphere. Ambient temperature and variation in sleep duration partially explained these results. Results are consistent across 23 countries.

## Methods
### Participants
This study is a retrospective analysis of data from 125,295 participants who bought, setup and registered to use an FDA-approved under-mattress sleep sensor for personal use (Withings Sleep Analyzer; WSA) between January 2020 and September 2023. Participants were included if they used their devices regularly, defined as ≥4 recordings per week and ≥28 valid AHI measurements per year (i.e., ≥5 hours sleep, see below), similar to previous studies[2,20,21]. Only participants with a yearly average AHI ≥ 5 events/h were included in this analysis. Participants were geo-localised to the closest, largest city in each time-zone within a country (in countries with multiple time-zones present) or the largest city if there was only a single time-zone within a country. More precise location was not available due to ethical considerations around privacy. All participants provided written consent through the Withings app for their deidentified data to be used for research purposes when signing up for a Withings account. The study was approved by the Flinders University Human Research Ethics Committee (Project number: 4291).

### Objective sleep monitoring
The Withings Sleep Analyser (WSA) is a non-wearable sleep monitoring device placed under the mattress that estimates the AHI and sleep stages. This is achieved via automated proprietary algorithms from a built-in microphone and ballistographic assessment of movement, heart rate, and respiratory motion from a pressure sensor[22]. To ensure consistent measurements across days, the WSA monitors pressure inside the inflated air bladder and periodically re-calibrate itself to ensure consistent pressure. The WSA-estimated AHI has good agreement with in-laboratory polysomnography-derived AHI to classify moderate-to-severe OSA (88% sensitivity and 88% specificity to detect ≥15 events/h sleep)[2,22]. The estimated AHI has minimal bias against in-laboratory polysomnography derived AHI when the AHI is considered as a continuous variable. Validation studies were conducted with the AASM 2012 criteria including a hypopnea definition with a reduction of flow above 30% of baseline and a 3% desaturation or an arousal[23]. The AHI cannot be reliably calculated for nights with a sleep duration of less than 5h[22].

Our main outcome of interest was change in AHI (referred to as "AHI change" throughout this manuscript) between a given night and yearly average, expressed as a percentage of the yearly AHI for each participant using the following equation:

$$cAHI_{d,y,p} = 100 * \frac{AHI_{d,y,p} - A\overline{H}I_{y,p}}{A\overline{H}I_{y,p}} \qquad (1)$$

Where $cAHI_{d,y,p}$ represents the change in AHI, in %, for a day $d$, a year $y$, and participant $p$ and $A\overline{H}I_{y,p}$ represents the averaged AHI for a given year $y$ and participant $p$. Similarly, we expressed change in total sleep time and sleep onset time in secondary analyses. We also used clinically defined OSA severity categories in secondary analyses (mild: 5-15; moderate: 15–30 and severe: >30 events/h).

### Seasonal and atmospheric variable assessments
We extracted atmospheric measurement from the fifth generation of European Reanalysis (ERA5) dataset[24] including hourly air temperature, measured at 2 m from the ground for each main city (500 km² around the main location) between 2020 and 2023. ERA5 is a climate model previously shown to provide a satisfactory proxy to station-based data series and used to assess the effect of ambient temperature on health[25]. Mean 24 h temperatures were calculated for each location. We also extracted hourly dew point temperature, total cloud cover, wind speed and surface pressure for each location, which were subsequently averaged over a 24 h period. Relative humidity was calculated using MetPy based on an existing formula[26]. These atmospheric variables were time-matched to each of the nightly sleep observations for each participant. We extracted measurements of fine particulate matter density with aerodynamic diameter of less <2.5 μm concentration from the European Centre for Medium-Range Weather Forecasts Atmospheric Composition Reanalysis 4 model as a measure of air pollution[27]. This model has shown good agreement against station-based measurements[28].

### Statistics and Reproducibility
We used non-linear fixed effect models with subject/year strata intercepts to account for potential year-to-year variation in AHI within individuals. For continuous outcomes we used the gaussian family with identity link function for the regression models. In some sensitivity analyses with a binary outcome, we used binomial family with logit link function. We modelled seasonal effects using natural splines of time (Julian day 1–365; with 8 degrees of freedom [df]) and further controlled for an indicator of day of the week in our baseline model. Since the day of the year is a construct variable that may encompass multiple different exposures variables including behavior (e.g., sleep, physical activity) and/or weather changes (e.g., temperature, cloud cover), we ran a fully adjusted model to investigate if seasonal effect persisted after adjustment for the following potential confounders: total cloud cover (4 df), relative humidity (4 df), air pollution (4 df), wind speed (4 df), surface pressure (4 df), yearly sleep duration irregularity (4 df), and yearly sleep timing irregularity (4 df). This analysis approach was first undertaken in different latitude categories (−90 to −30°, 30 to 0°, 0 to 30° and 30 to 90°) and then in countries with at least 200 users. Secondary analyses examined specific subgroups of interest including age (10 y bins), sex, habitual sleep duration categories ( < 6, 6 to 7, 7 to 8, 8 to 9 and 9+ hours) and OSA severity categories (mild, moderate and severe). Estimated marginal means and 95%CI are reported in all figures.

To further explore the potential associations between environmental variables and percentage AHI change, we constructed two additional models per variable of interest. These predictors included: 24 h average temperature, total cloud cover, relative humidity, wind speed, air pollution, surface pressure, sleep duration irregularity, and sleep timing irregularity. The first model for each predictor was adjusted for the day of the year and the day of the week, and the second model was adjusted for additional potential confounders. We compared exposure-response curves between the minimally adjusted models and the fully adjusted models to assess collinearity between variables. Each exposure of interest had different potential confounders in the fully adjusted model (see supplementary Table 1 for a detailed list). The effects were summarized using estimated marginal means, comparing AHI change at the 75th percentiles vs. 25th percentile of the exposure variable. In case of U- or J-shape association, we compared the estimated marginal means at the 95th vs. 50th percentiles, and the 5th vs. 50th percentiles. The models were implemented in the R

programming language[29] (version 4.3.3), using the dlnm[30] and gnm packages[31]. Model specifications can be found in the supplementary code section of the supplementary material.

### Sensitivity and supplementary analyses

To further validate our findings, we conducted several sensitivity and supplementary analyses. Firstly, we reproduced the analysis only in data after September 2022, a period where COVID19 was less likely to confound the observed results. Secondly, we reproduced the main analysis by varying the degree of freedom for the seasonal splines to 4, 6 and 8. We also reproduced the main analysis using absolute values of AHI as an outcome within each OSA severity category. Finally, we reproduced the main analysis by estimating the probability of OSA status (moderate to severe: AHI ≥ 15; severe OSA: AHI ≥ 30) across the year.

### Reporting summary

Further information on research design is available in the Nature Portfolio Reporting Summary linked to this article.

## Results

### Participants characteristics

There were 70,052 participants with at least mild OSA available for the final analyses (see flowchart in Supplementary Fig. 1). The demographics of the final sample are shown in Table 1. The geographical locations of participants are shown in Fig. 1, and a total of 23 countries with ≥200 participants were included. There were 66,686 participants (95.2%) in the 30 to 90⁰ latitudes, 1070 (1.5%) in the 0 to 30⁰ latitudes, 556 (0.8%) in the −30 to 0⁰ latitudes, and 1740 participants (2.5%) in the −90 to −30⁰ latitudes. On average, 492 (± 341) nightly recordings were available per participant, with a total of ~34 million nightly recordings. The average overall coefficient of variation for AHI (SD/mean) was ~51%, ~69% for mild OSA, 52% for moderate OSA, and 37% in the severe OSA group.

### Seasonal variation in OSA severity

We observed seasonal variation in AHI. The AHI was ~5% higher during summer and winter in the northern hemisphere compared to spring/autumn, and 10-15% higher during summer in the southern hemisphere (Fig. 2a) compared to autumn. However, the pattern of seasonal effects on AHI change was variable even within the same latitude categories. For example, spring and summer in Japan were associated with a 10% increase in AHI, relative to autumn and winter whereas a higher AHI in the United States for this same period was not detected. Similarly, in some European countries such as France, there was no evidence of exacerbation of OSA during spring or autumn, but summer (July/August) and winter

(December/Jan) was associated with a 5% increase in AHI compared to November or March. In Australia, the pattern was also different compared to Europe, but similar to Brazil, whereby AHI was lowest from February to April, but steadily increased and reached its maximum ( + 10 to 15%) during the summer months (December and January). Seasonal variation in AHI was greater in men than women (Fig. 3a). Seasonal variation was also higher in mild vs. moderate and severe OSA (Fig. 3b), and in participants <60 vs. ≥60 years old (Fig. 3d). Seasonal variation was higher in participants with a habitual sleep duration below 8 h vs. participants who slept on average more than 8 h (Fig. 3d).

The exposure-response curve between the day of the year and the AHI change was similar to the main analysis when data was restricted to before or after September 2022 (Supplementary Fig. 2). The exposure-response curve was also similar when more restrictive degrees of freedom for the seasonal spline were used (Supplementary Fig. 3). The seasonal changes in absolute AHI were consistent in shape between different OSA categories (Supplementary Fig. 4) compared to the main analysis. The absolute changes were, however, greater in moderate to severe OSA and severe OSA vs mild OSA (Supplementary Fig. 4). While the changes in absolute AHI at the population may appear small ( < 2 events/hour), the odds of presenting with moderate to severe OSA (peak to through differences, see Supplementary Fig. 5) or severe OSA was increased by mean [95%CI]; 15.3 [14.7, 16.0] % and 18.7 [17.9, 19.6] % seasonally, respectively.

### Environmental factors explaining seasonal variation in OSA severity

There was an association between multiple environmental and sleep factors with AHI change (see Fig. 4). There were minimal correlations between environmental and sleep variables (Supplementary Fig. 6). There was a non-linear association between AHI change and temperature, such that higher temperatures (25th vs. 75th percentiles; 6 vs. 18⁰C) were associated with a 6.4% (95%CI; 6.3–6.5%) increase in AHI. Similarly, there was an association between surface pressure and AHI change, where lower atmospheric pressure was associated with higher AHI. We did not find an association of either outdoor air pollution or outdoor relative humidity with AHI (Fig. 2c, d). Days with higher wind speeds were also associated with higher AHI (75th vs. 25th; 1.7 [1.6, 1.8]%).

Variation in total sleep time was associated with a higher AHI, where a 132-minute increase (50th vs. 95th) in sleep duration compared to the yearly average was associated with a 5.8% (5.7 to 5.9%) increase in AHI. Short sleep duration compared to the yearly average (5th vs. 50th; 94 minutes reduction) was also associated with a modest increased AHI (0.9 [0.8, 1.0] % increase). Variation in sleep timing (time in bed) was associated with AHI (50 vs 95th:

## Table 1 | Population sample demographics

| | | Overall | Tertiles of OSA variability (AHI SD, events/h) | | |
| --- | --- | --- | --- | --- | --- |
| | | | T1 2.5 to 6.3 | T2 6.3 to 9.8 | T3 ≥9.8 |
| n | | 70,052 | 23,351 | 23,350 | 23,351 |
| Age (years) | | 53 (13) | 49 (12) | 54 (13) | 56 (13) |
| Sex n (%) | Men | 57,007 (81.4) | 17,863 (76.5) | 18,945 (81.1) | 20,199 (86.5) |
| | Women | 13,045 (18.6) | 5488 (23.5) | 4405 (18.9) | 3152 (13.5) |
| BMI (kg/m²) | | 28.8 (5.7) | 27.2 (5.1) | 28.6 (5.3) | 30.8 (6.1) |
| Mean number of nights per person | | 492 (341) | 428 (321) | 518 (345) | 531 (348) |
| Mean AHI (events/h) | | 18.0 (14.0) | 8.3 (2.7) | 14.5 (7.2) | 31.0 (16.1) |
| SD AHI (events/h) | | 9.1 (4.8) | 5.0 (0.8) | 7.8 (1.0) | 14.5 (4.6) |
| Mean sleep duration (h) | | 7.3 (0.9) | 7.5 (0.8) | 7.4 (0.9) | 7.1 (1.1) |
| SD sleep duration (min) | | 74.7 (42.7) | 69.7 (35.0) | 73.5 (44.9) | 80.9 (46.5) |

*AHI* Apnea-hypopnoea-hypopnea index, *BMI* body mass index.

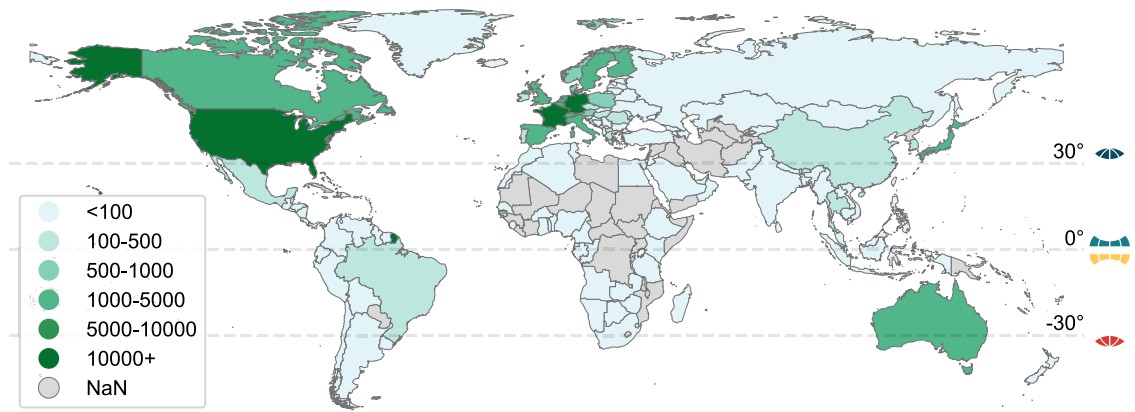

**Fig. 1 | Geographical locations of the participants.** Worldwide distribution of study participants; legend indicates the number per country.

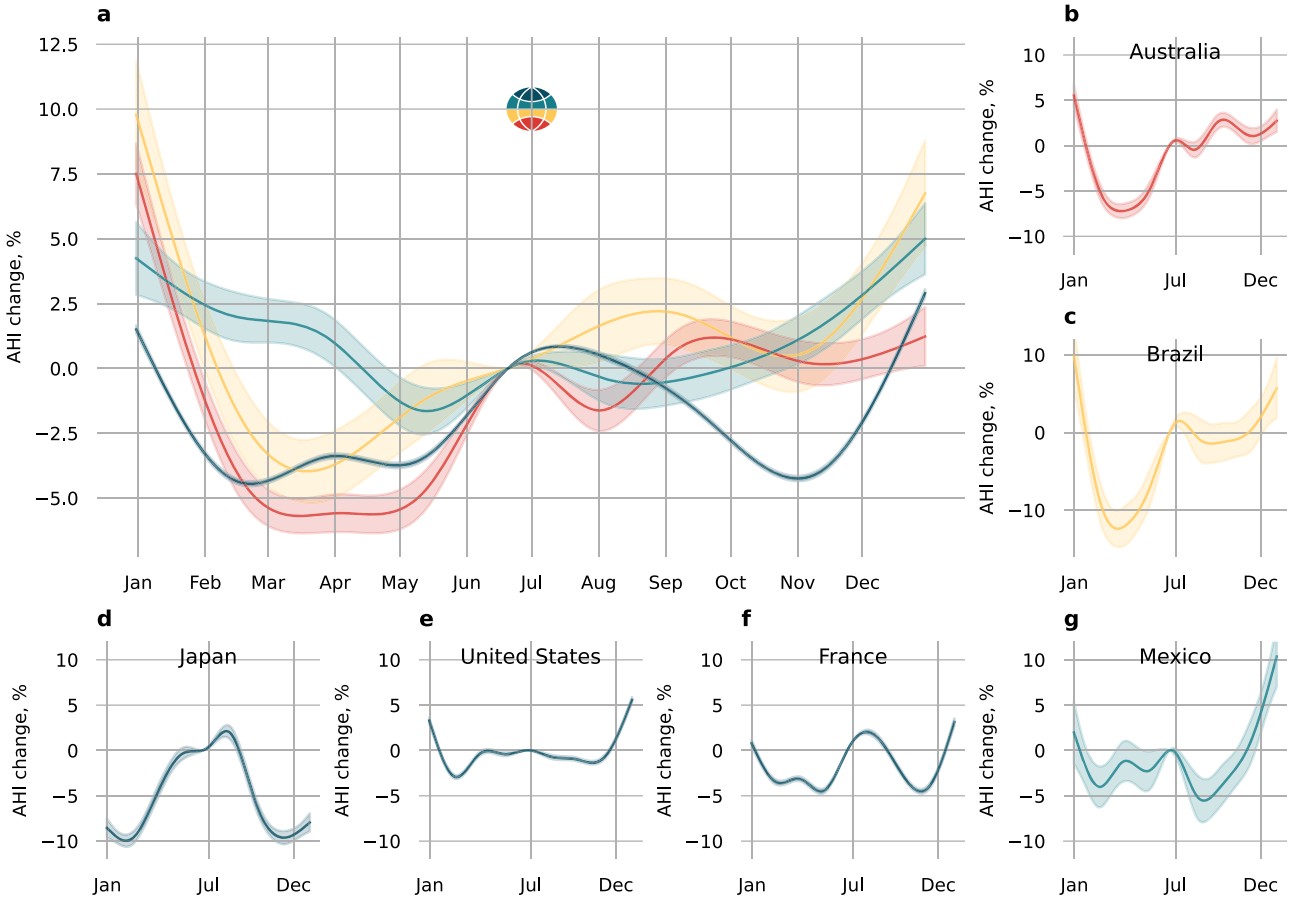

**Fig. 2 | Seasonal variation in the apnea-hypopnoea index. a** Variation in apnea-hypopnea index (AHI) for different latitude categories across seasons, using the 21st of June as the reference (summer and winter solstice in the northern and southern hemisphere, respectively). **b–g** Seasonal variation in AHI for different countries.

Country by country number of users and nights is available in Table 2. Number of users per latitude categories, accounting for participants that have moved locations between year, are as follow: 90° to 30°, N = 67,526; 30° to 0°, N = 1,640; −30° to 0°, N = 1038; −30° to −90°, N = 1640. Source data are available with this paper.

−6 vs. 112 min; 1.6% [1.5–1.7%]) (Fig. 4). The associations were similar in unadjusted vs. fully adjusted models suggesting that multi-collinearity between environmental variables was not an issue. Earlier time in bed compared to the yearly average was associated with AHI change in the unadjusted models but not in the fully adjusted models (see supplementary Supplementary Fig. 7). Indeed, once adjusting for variation in total sleep time, the association between earlier than normal time-in-bed with AHI change disappear (Supplementary Fig. 7 red vs black curve). This suggests that there was multi-collinearity between variations in sleep timing and variation in sleep duration, similar to a previous study[20].

**Country by country models**

The effect of environmental and sleep factors on AHI was consistent across most countries (see Table 2 for summary statistics and Supplementary Supplementary Fig. C1 to C12 for the exposure-response curves for the top 12 countries with the most users). High temperatures (75th vs. 25th) were associated with a 6–14% increase in AHI in Europe (except for Austria), but only a~2% increase in Australia and the US. Longer than usual TST (95th vs. 50th) was also consistently associated with a 3–11% increase in AHI in all countries. However, shorter than usual TST (5th vs. 50th) was not consistently associated with AHI (see Supplementary Table 2) in all countries. The effect

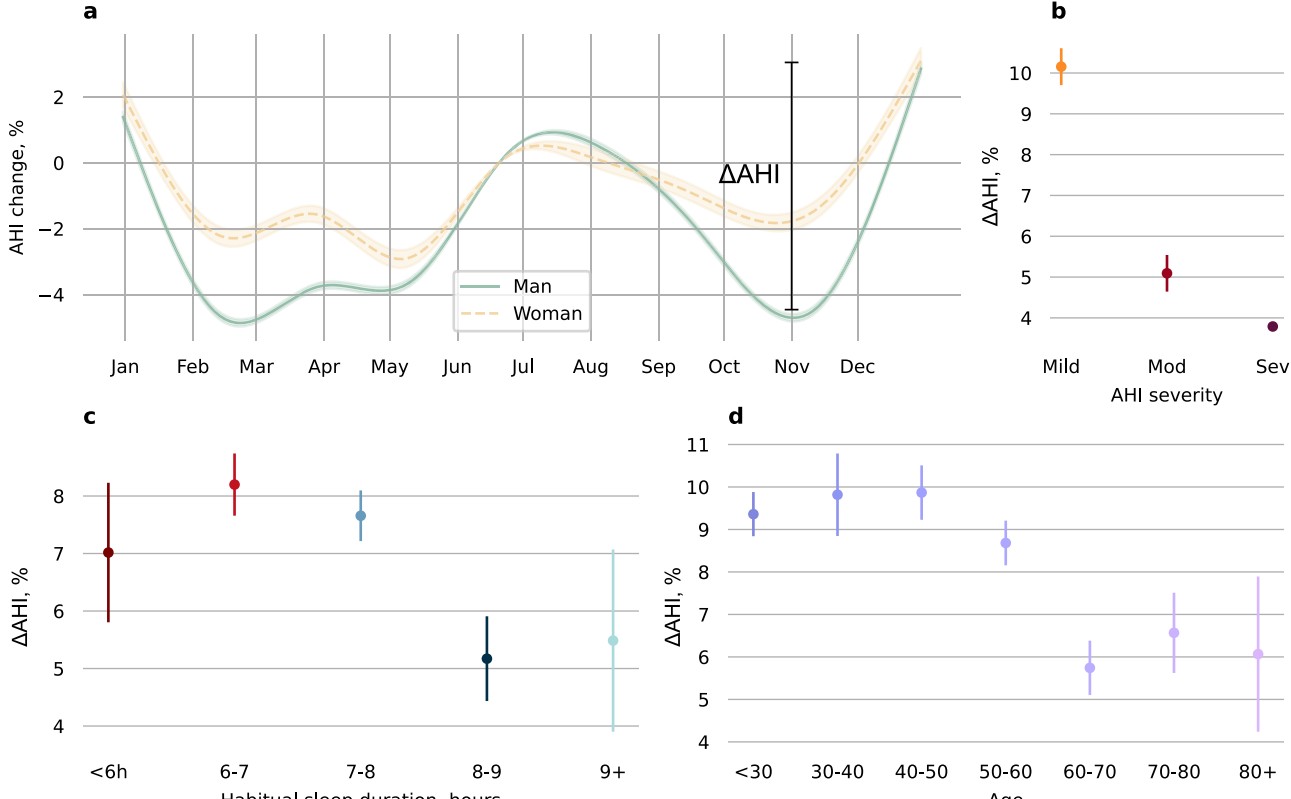

**Fig. 3 | Seasonal variation in the apnea-hypopnoea index for different subgroups.** Subgroups included (**a**) sex; (**b**) obstructive sleep apnea (OSA) severity categories based on standard apnea-hypopnoea index (AHI) cutoffs: 5 to 15 (mild), 15 to 30 (mod) and ≥ 30 events/h (sev); (**c**) habitual sleep duration categories; and (**d**) age categories. Analysis performed for the 30 to 90⁰ latitude category only (representing 95.2% of the dataset). Seasonal effect is summarized using the mean (95%CI) difference between the peak and the trough of the seasonal variation in OSA severity, referred to as ΔAHI. Number of participants included in the analysis: sex (Women: 12556, Men: 54970), OSA severity categories (see Supplementary Table 3), habitual sleep duration (<6 h: 10899, 6–7 h:28787, 7–8 h: 34352, 8–9 h: 12529, > 9 h:2736) and age (18–30: 2304, 30–40:9832, 40–50:16846, 50–60:19751, 60–70:12617, 70–80:5203, 80 + :1423). Source data are available with this paper.

of variation in surface pressure ( < 10–30 hPa) on AHI was <4% in most countries. Once accounting for environmental and sleep factors, the seasonal effect in AHI persisted, such that AHI was 10 to 20% greater on peak vs. trough months (Table 2 and see Supplementary Figs. SC1 to SC12 for the seasonal shape). In most countries, once models were adjusted for environmental variables, the seasonal component of AHI only was evidenced as a winter peak.

## Discussion

This large global study investigated seasonal patterns in OSA severity and found that the AHI was higher (8 to 19%) during winter and summer compared to spring and autumn in the northern hemisphere. The summer peak is likely to be partly explained by increased ambient temperatures, which were associated with a 2 to 15% increase in AHI severity consistently across the 23 studied countries. We also found that sleeping longer than the yearly average was associated with a further 3 to 10% increase in AHI severity, which may partially account for the peak in winter. Together, these findings have important implications for the diagnosis and long-term management of OSA and global sleep health implications.

There are several factors that may influence night-to-night variability in OSA severity. These factors include body/head position during sleep, circadian effects, sleep stage distributions and timing, changes in non-anatomical OSA endotypes, nasal resistance, and behavioural and lifestyle factors such as nutrition, physical activity, social activities, alcohol, caffeine use, tobacco intake, and medication use[5,32,33]. Some factors, such as weight gain, and physical activity and alcohol consumption, vary across seasons[22,34,35], which may further explain some of the observed seasonal variation in OSA severity. Indeed, adiposity is higher, and physical activity is lower during winter vs.

summer/spring[22,34], which is consistent with the corresponding increased AHI in the current study. Depressive symptoms, anxiety and insomnia symptoms may also be higher during winter months[36], which could lead to lighter sleep and thus, elevated OSA severity during winter[37]. Other potential explanations for increased AHI during winter may be increased nasal resistance due to respiratory illnesses such as the flux or COVID-19.

During winter, people tend to sleep ~20 min longer which may be explained by extended wake-up time (25 min later compared to summer) rather than earlier bedtime[38]. Hence, sleep opportunity is mostly extended in the early morning, a period where REM sleep is more likely to occur[39] and to increase the AHI[37]. Extended sleep durations may also increase the proportion of light sleep, which could also increase the AHI[37]. This may explain, at least in part, increased OSA severity due to increased sleep duration compared to the yearly average observed in this study; however robust nightly analyses with sleep staging are required to confirm this hypothesis. Sleep was shorter during summer months; hence, extended REM sleep periods are unlikely to explain the summer peak in OSA severity. Some studies suggest higher temperatures reduce sleep duration and cause poorer self-reported sleep quality[40–43], hence, AHI may be higher on hot nights due to lighter sleep. This is consistent with our observations that AHI was higher with increased temperatures.

Taken together, all available potential sources of variability investigated in the current study (i.e., seasonal, environmental, temperature and sleep-effects) remained modest and suggest that other factors must play a key influential role in mediating night-to-night variability in OSA severity. Ambient temperature and seasonal variation in sleep duration were the two main factors influencing seasonal variation in AHI. Indeed, while present, the influence of other factors such as high wind speed, surface pressure, air

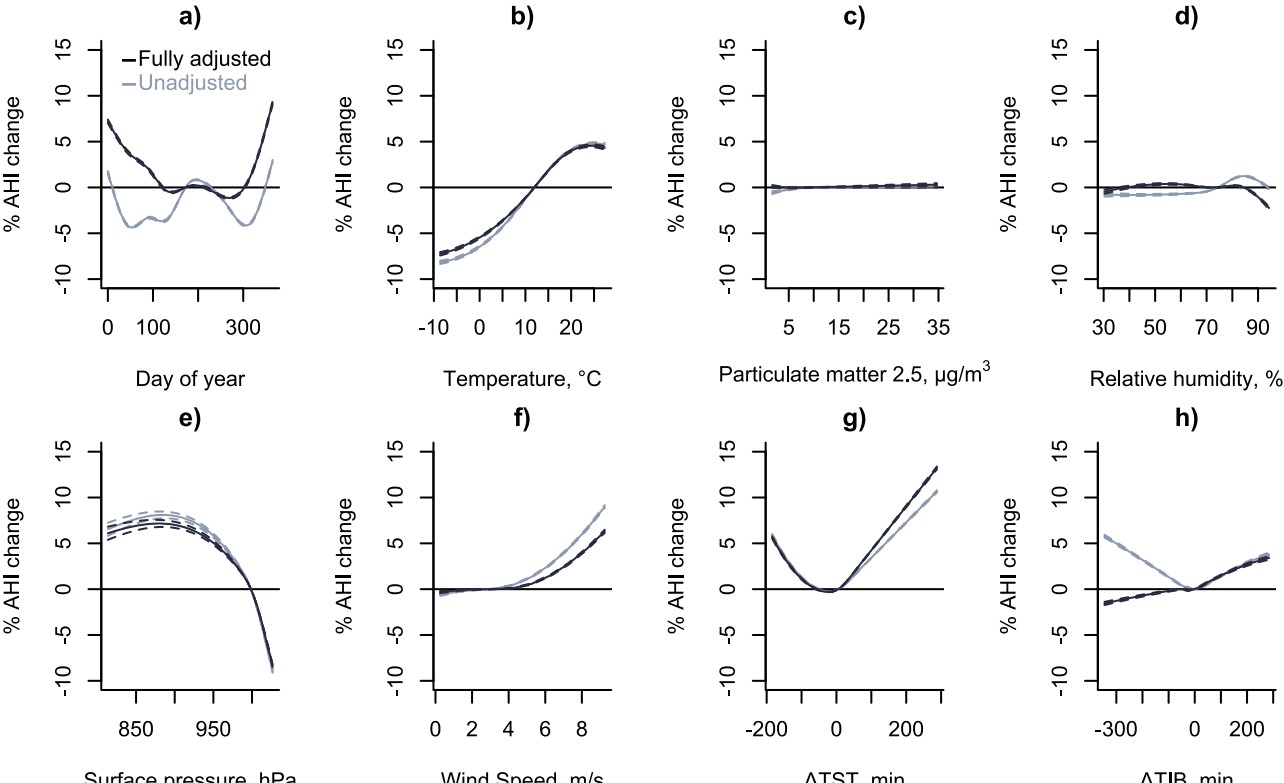

**Fig. 4 | Associations of different environmental and sleep-specific factors with seasonal variation in the apnea-hypopnoea-index.** Variation in the apnea-hypopnoea-index (AHI) for unadjusted (blue) and fully adjusted models (black). Notes that the associations are similar in unadjusted vs. fully adjusted models suggesting that multi-collinearity between environmental variables is not an issue (**a**) Day of the year (21st of June as reference), (**b**) 24 h average temperature, (**c**) density of particulate matter with diameter of less than 2.5 um (**d**) relative humidity **e**) surface pressure, (**f**) wind speed (**g**) difference between a given night's total sleep time (TST) with the yearly TST average (ΔTST) in minutes and (**h**) similar to (**g**) but for time in bed (ΔTIB) – see Supplementary Fig. 7 for extra analyses on potential collinearity with ΔTST. All graphs represent estimated marginal means using the 50th percentiles as the reference value (except for **a**), and the x-axis limits were set as the 1st percentile and the 99th percentiles. Source data are available with this paper.

pollution, and variable sleep timing on AHI variability were small. Nonetheless, a seasonal effect that accounts for ~20% of the variation in AHI is meaningful. Some pharmacological trials in OSA patients only show modest reductions in OSA severity, such as a 10 to 20% reduction[44]. Similarly, meta-analyses of OSA severity reduction in response to diet and exercise have demonstrated ~20% reductions in AHI[45]. Thus, our results highlight the importance of reporting data collection months in clinical trials and time-matching OSA severity assessment for intervention and control participants in trials. Therefore, while the observed effect is unlikely to be clinically meaningful at an individual level, at the population level it may lead to misestimation of OSA severity which could have implications for clinical trials. There may also be implications in term of safety, since the odds of having moderate to severe OSA varies by 15% seasonally, with a peak during summer and winter compared to spring or autumn. Our findings also further question the use of single-night sleep studies to diagnose OSA since many factors (e.g., temperature, sleep schedules, seasons) vary over time and have an impact on OSA severity, which may partially explain the high misdiagnosis rates in OSA using single night studies[5].

There are some limitations with our study which should be considered when interpreting the findings. Clinical information regarding co-morbid conditions, sleepiness, and treatment status were unavailable. Some behavioural information such as alcohol intake and smoking status that could have helped understand the mechanisms behind a seasonal pattern of OSA were also unavailable. Information on participants socio-demographics was also unavailable. Similarly, the lack of more precise geographical locations than nearest major city may have limited our ability to detect some associations with environmental variables. In addition, we had to rely on outdoor ambient temperature, and we had no information on air conditioning/

heating usage. This may have confounded some of the associations at the country level. For example, the association between high temperature and AHI had a lower effect size in the United States compared to other countries, which is also one of the countries with the largest usage of air conditioning worldwide[46]. The above limitations, and previous studies showing correlation between air pollution and temperature[47], or sleep duration and timing[20], may have reduced our ability to determine associations between environmental and sleep -related variables and changes in AHI. Our population sample is also biased towards men, and the participants were likely of higher socio-economic status since they could afford to purchase an unsubsidised sleep tracking device. Most participants also resided in highly developed countries, so they may have also had access to more favourable sleeping environments and heat stress-mitigation strategies such as air conditioning[46]. This high socio-economic bias is common in the sleep research literature[48]. Our results highlight the urgent need for global strategies to collect appropriate sleep and temperature data worldwide[48]. Our results remained consistent once data were restricted to post September 2022, suggesting that the COVID-19 pandemic did not seem to influence our main results. However, we did not have access to clinical symptoms and potential COVID-19 infections and/or long COVID, which could have influenced the overall seasonal variability of OSA severity. Our study also has several strengths, including a longitudinal design with >1.5 years of nightly recordings for most participants, a much wider geographical coverage than any previous studies, and a dataset larger than prior work. Furthermore, we reproduced our estimates across multiple countries, which supported consistency of key results.

The estimated AHI from the under- mattress sensor includes fewer input variables in which to detect respiratory events compared to

**Table 2 | Association between nightly AHI, expressed as a % of yearly AHI, and select seasonal, environmental and sleep variables for different countries**

| Country | N nights (x1000) | N users | Seasonal AHI change* (peak vs. trough) | ΔTST (95th vs. 50th) | T°C (75th vs. 25th) | Pressure (75th vs. 25th) |
|---|---|---|---|---|---|---|
| Netherlands | 836 | 1631 | 20.7 (19.3, 22.1) | 9.8 (9.4, 10.3) | 14.7 (13.8, 15.6) | 5.7 (5.4, 6.1) |
| Denmark | 315 | 638 | 20.2 (17.5, 23.0) | 9.6 (8.9, 10.3) | 11.8 (9.9, 13.7) | 3.7 (3.1, 4.2) |
| Ireland | 136 | 273 | 19.1 (15.9, 22.4) | 5.9 (4.8, 7.0) | 6.3 (4.3, 8.2) | 6.0 (5.2, 6.9) |
| United Kingdom | 2425 | 4690 | 17.1 (16.3, 17.9) | 7.0 (6.8, 7.3) | 9.0 (8.5, 9.5) | 3.5 (3.3, 3.7) |
| Hungary | 134 | 267 | 14.0 (10.3, 17.7) | 5.3 (4.2, 6.5) | 8.9 (6.2, 11.6) | 2.8 (2.2, 3.5) |
| Romania | 95 | 241 | 13.9 (9.1, 18.8) | 4.1 (2.6, 5.6) | 10.3 (6.8, 13.8) | 2.7 (1.9, 3.6) |
| Japan | 908 | 1884 | 13.9 (12.2, 15.6) | 5.4 (4.9, 5.8) | 13.5 (12.0, 14.9) | 2.9 (2.6, 3.2) |
| Norway | 257 | 560 | 13.1 (10.1, 16.2) | 14.6 (13.7, 15.4) | 9.4 (7.4, 11.3) | 3.2 (2.5, 3.8) |
| Germany | 7489 | 13738 | 13.0 (12.6, 13.5) | 9.0 (8.9, 9.2) | 8.3 (8.0, 8.6) | 3.4 (3.3, 3.5) |
| Italy | 637 | 1251 | 13.0 (11.3, 14.7) | 6.1 (5.6, 6.6) | 7.3 (6.1, 8.5) | 1.3 (1.0, 1.7) |
| Portugal | 177 | 363 | 12.7 (9.9, 15.5) | 7.2 (6.2, 8.2) | 4.5 (2.8, 6.1) | 1.1 (0.4, 1.8) |
| Australia | 632 | 1879 | 12.7 (11.3, 14.0) | 7.7 (7.2, 8.2) | 2.0 (1.4, 2.7) | 2.9 (2.4, 3.3) |
| Belgium | 530 | 992 | 12.6 (10.9, 14.3) | 8.5 (8.0, 9.1) | 8.9 (7.8, 10.0) | 2.9 (2.5, 3.3) |
| Poland | 231 | 512 | 11.9 (9.1, 14.6) | 6.1 (5.3, 7.0) | 10.7 (8.8, 12.7) | 3.6 (3.0, 4.2) |
| Sweden | 533 | 1066 | 11.9 (9.6, 14.1) | 11.5 (11.0, 12.1) | 6.6 (5.0, 8.3) | 4.4 (4.0, 4.8) |
| Czech Republic | 122 | 264 | 11.3 (7.5, 15.2) | 8.9 (7.6, 10.1) | 3.4 (0.8, 6.0) | 3.7 (2.7, 4.6) |
| Switzerland | 1436 | 2740 | 10.6 (9.5, 11.7) | 9.8 (9.4, 10.1) | 9.3 (8.6, 10.0) | 1.5 (1.3, 1.6) |
| Spain | 425 | 999 | 10.3 (8.4, 12.1) | 4.8 (4.1, 5.4) | 5.8 (4.7, 7.0) | 0.4 (−0.1, 0.9) |
| France | 5324 | 10816 | 10.0 (9.4, 10.5) | 5.9 (5.7, 6.1) | 6.9 (6.5, 7.2) | 3.2 (3.1, 3.3) |
| Finland | 566 | 1162 | 9.9 (7.7, 12.2) | 8.0 (7.4, 8.5) | 8.8 (7.1, 10.4) | 4.3 (3.8, 4.7) |
| Austria | 457 | 836 | 9.0 (7.3, 10.8) | 8.8 (8.2, 9.5) | 0.4 (−0.8, 1.5) | 0.2 (−0.2, 0.5) |
| United States | 8726 | 18473 | 8.7 (8.3, 9.1) | 3.3 (3.2, 3.4) | 2.2 (1.9, 2.5) | 2.2 (2.0, 2.5) |
| Canada | 806 | 1700 | 7.5 (6.2, 8.8) | 4.8 (4.3, 5.2) | 3.8 (2.9, 4.7) | 3.0 (2.3, 3.7) |

*Seasonal effect is summarized using the mean (95%CI) difference between the peak and the trough of the seasonal variation in OSA severity. ΔTST variation in sleep duration from the yearly mean. T°C temperature in degrees Celsius.

conventional polysomnography. Accordingly, other physiological aspects of OSA (e.g., hypoxia, posture) as well as event-specific characteristics such as ventilatory burden, ventilatory drive, central vs obstructive events cannot be measured. However, validation studies in over 150 participants support the device performance characteristics with minimal bias versus polysomnography AHI[2,49]. The classification of OSA severity based on the estimated AHI ability is also ~80% accurate (mild-to-severe OSA: 89% sensitivity and 75% specificity; moderate-to-severe OSA: 88% sensitivity and 88% specificity; severe OSA: 86% sensitivity and 91% specificity)[22]. The device performance is similar to other wearables and nearable devices in estimating OSA severity such as mandibular movement-based sensors[50], chest-patch type of sensors[51], forehead devices[52], finger-worn oximetry devices[53,54], other mattress sensors[22,55], RADAR based technologies[56]. Furthermore, OSA prevalence estimates using non-contact multi-night data yield very similar findings to previously published literature[1,2]. Similarly, misclassification rates and AHI variability are also comparable to data derived from multi-night in-laboratory polysomnography and other home sleep apnea tests[2,6,7]. Thus, these findings provide support that the multi-night mean AHI estimates derived in the current study provide comparable insight to gold-standard

polysomnography AHI while being less cumbersome and allowing nightly data collection over ~1.5 years per individual.

## Conclusions

Using data collected with an under-mattress sleep sensor over a 3.5-year period, we observed a seasonal effect on OSA severity, with sleep duration extension and ambient temperatures explaining some of these associations. These findings highlight that decision-making based on brief recording periods may not represent longer-term trends in OSA severity. They also suggest the importance of time-matched controls in OSA treatment studies and the need to report data collection months in OSA clinical trials. Further studies to determine the physiological mechanisms underpinning the seasonal variation in OSA severity are required.

## Data availability

The dataset associated with this study is stored in a proprietary repository (Withings) and cannot be shared publicly due to concern for privacy, ethical and legal reasons. The investigator team accessed the data through an application process to Withings, designed to safeguard user confidentiality, as outlined in the terms and conditions and privacy policy documentation. Queries for data access can be directed to Withings (contact-sup@withings.com) with a timeframe for response of four weeks. Specific de-identified raw data that support the findings of this study, including individual data, are available from the corresponding author (bastien.lechat@flinders.edu.au) upon request subject to ethical and data custodian (Withings) approval described above. The timeframe for response to requests will be four weeks. ERA5 weather data and climate model projections are freely available from the Copernicus data store (https://cds.climate.copernicus.eu/)[57]. Source data to reproduce Figs. 2 and 3 are available in the Supplemental Data 1 file. Exposure-response curve and Source data to reproduce Fig. 4 is provided in the Supplemental Data 2 file.

## Code availability

No specific code was developed as part of this study. Model specification for the statistical analysis is presented in supplementary material.

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

## Acknowledgements

D.J.E. is supported by a National Health and Medical Research Council (NHMRC) of Australia Leadership Fellowship (1196261). B.L. is supported by a NHMRC of Australia Emerging Leadership Fellowship (2025886). JLP is supported by the French National Research Agency in the framework of the "e-health and integrated care and trajectories medicine and MIAI artificial intelligence (ANR-19-P3IA-0003)" Chairs of excellence from Grenoble Alpes University.

## Author contributions

B.L., H.S., D.P.N., P.C. and D.J.E. developed the study concepts and aims. B.L., D.P.N. and J.M. performed the data extraction. B.L., K.S., A.C.R., & D.P.N. performed data analyses. BL drafted the manuscript. B.L., D.P.N., K.S., L.P., H.S., A.R., A.V., J.M., R.J.A., J.L.P., P.E., P.C., and D.J.E. provided important insight on data analysis, interpretation and contributed to drafting and to the final version of the manuscript.

## Competing interest

The authors declare the following competing interests: PE serves as a consultant for Withings. Outside the submitted work, BL has had research grants from Withings, the NHMRC and the Medical Research Future Fund (MRFF). DJE has had research grants from Bayer, Apnimed, Takeda, Invicta Medical, Eli Lilly and Withings. DJE currently serves as a scientific advisor/consultant for Apnimed, Invicta Medical, SleepRes, Takeda and Mosanna. ACR has received research funding from the Lifetime Support Authority, Sleep Health Foundation, Flinders Foundation, Compumedics, and Sydney Trains, and speaker and consultancy fees from Teva Pharmaceuticals, Sealy Australia, and the Sleep Health Foundation for work unrelated to this study. HS reports consultancy and/or research support from Re-Time Pty Ltd, Compumedics Ltd, the American Academy of Sleep Medicine Foundation, and Flinders University. JLP JLP has received lecture fees or conference travel grants from ResMed, Philips, Agiradom and Bioprojet, and has received unrestricted research funding from ResMed, Bioprojet, Fondation de la Recherche Medicale (Foundation for Medical Research), Direction de la Recherche Clinique du CHU de Grenoble (Research Branch Clinic CHU de Grenoble), and fond de dotation "Agir pour les Maladies Chroniques" (endowment fund "Acting for Chronic Diseases"). AV has received competitive research funding and equipment from ResMed and Philips Respironics for research unrelated to and outside the submitted work. PC reports grants from NHMRC, Medical Research Future Fund, Flinders Foundation, Invicta Medical, Garnett Passe and Rodney Williams Memorial Foundation, Defence Science and Technology Group. None of the other authors have any potential conflicts to declare.
