## [Transparent Peer Review file · Communications Medicine]

OSA severity varies by season and environmental influences such as ambient temperature

Corresponding Author: Dr Bastien Lechat

Version 0:

Reviewer comments:

Reviewer #1

(Remarks to the Author)

Overall Assessment:

This study presents an extensive exploratory analysis with potentially intriguing associations among seasonal pattern and environmental factors. However, the methodological framework requires refinement to ensure analytical transparency and reproducibility.

Major Concerns:

1. (Line 114) The methodological description of the nonlinear mixed-effects model implementation remains ambiguous. Given that the cited gnm package primarily estimates fixed effects, please provide the complete model specification code in R

Minor Concerns:

1. Data Reporting:

- (Line 74) Clarify the geospatial resolution of participant locations (individual addresses vs. aggregated regional data)

2. Statistical Specification:

- (Line 114) Explicitly state the distribution family (Poisson/Gaussian) and link function for the regression model

- (Line 158) Consider constraining the temporal smoothness parameters through:

To reduce degrees of freedom for seasonal splines or choose best df using AIC/BIC criteria

- (Line 171) Provide mathematical formulation for Δ AHI calculation

3. Analytical Transparency:

- (Line 149) Correct typographical error ("30-")

4. Data Visualization:

- (Figure 4) Implement dual enhancements:

a) Annotate the chromatic scheme with a descriptive legend

b) Optimize y-axis scaling through variable-specific ranges rather than uniform limits

Reviewer #2

(Remarks to the Author)

This study examines the seasonal variations in the severity of obstructive sleep apnea (OSA) using large-scale data from Withings Sleep Analyzer users, assessing the impact of environmental factors such as temperature, humidity, and air pollution on the apnea-hypopnea index (AHI). The findings provide valuable insights into seasonal influences on OSA and may have implications for clinical management. However, several critical issues need further clarification and improvement to enhance the study's scientific rigor and applicability.

1. Clinical Significance and Statistical Representation of AHI Changes

1) The study reports a 5-15% seasonal variation in AHI but does not provide actual values, which may lead to misinterpretation. For example, given that the average AHI is 18, this corresponds to a change of 0.9 to 2.9 events/h.

Providing these absolute values would help contextualize the findings.

2) Percentage changes in AHI may exaggerate seasonal effects in mild OSA while underestimating them in severe OSA, as the relative impact differs depending on baseline severity.

3) I suggest using standardized AHI metrics (e.g., Z-score) to ensure comparability across OSA severity groups and provide a more intuitive measure of AHI variability.

2. Selection Bias in the Study Population: T The study is based on Withings Sleep Analyzer users, who are likely to be more health-conscious and of higher socioeconomic status, potentially leading to an underestimation of OSA prevalence and severity in lower-income populations. The limitations section should explicitly acknowledge this selection bias and discuss its implications for the generalizability of the findings. If data are available, I suggest analyzing AHI seasonal variation across

different socioeconomic groups to assess potential disparities.

3. Collinearity Among Environmental Variables: The study assesses the impact of temperature, humidity, and air pollution on AHI, but these variables are likely highly correlated, which may introduce collinearity issues in the analysis. I recommend using Principal Component Analysis (PCA) or LASSO regression to reduce collinearity and identify the most influential environmental factors.

4. Additionally, interactions between temperature, humidity, and air pollution should be explored.

5. Impact of COVID-19 on AHI Variability: T The data collection period (2020–2023) coincides with the global COVID-19 pandemic, but the study does not discuss how the pandemic may have influenced AHI.

1) The pandemic increased the risk of OSA through multiple mechanisms. Beyond direct infection effects, pandemic-related lifestyle changes, including weight gain, reduced physical activity, psychological stress, and limited healthcare access, may have exacerbated OSA severity and could confound the observed seasonal effects.

2) COVID-19 itself exhibits strong seasonal patterns, with peaks often occurring in colder months and coinciding with major holidays when social gatherings increase. These seasonal surges in infection rates may contribute to changes in sleep quality, respiratory health, and indirectly, OSA severity. This factor should be considered in interpreting the study's seasonal findings.

3) "Long COVID" could have lasting effects on respiratory function, potentially influencing AHI variability independently of seasonal changes.

6. The manuscript does not explain why the thresholds of ≥ 4 nights per week and ≥ 28 nights per year were chosen as inclusion criteria.

Version 1:

Reviewer comments:

Reviewer #1

(Remarks to the Author)

The authors have addressed all the questions I mentioned. I don't have any comments.

Reviewer #2

(Remarks to the Author)

I appreciate all the revisions and the additional sensitivity analyses the authors have provided. However, there are still several issues that should be addressed to improve clarity and consistency:

1. In the first Supplementary Figure S5, the y-axis is titled "AHI (events/hour)", which could be misleading. From the visual pattern, the values appear to represent Δ AHI (events/hour), rather than the actual absolute AHI values. The authors should clarify this in both the figure caption and axis labels.

In the right figure of Supplementary Figure S5, it is unclear why the green curve corresponds to the AHI range (-1, 15]. If this is meant to group together participants with no OSA and mild OSA, the lower bound should begin at 0, not -1. Furthermore, the cut-off points at 15 and 30 are standard clinical thresholds for moderate and severe OSA, respectively, yet the figure legend appears to assign 15 and 30 as thresholds for mild and moderate OSA. I suggest revisiting the category labels to align with standard clinical definitions and rerun the analysis if you misclassified them. Additionally, in the stratified analysis by OSA severity, it would be helpful to report the sample size, mean AHI, and range of AHI change for each group. I suggest including a summary table in the supplementary material to improve clarity and transparency. This would also aid in interpreting the following figure (Figure S6).

2. The second figure in the supplementary material is also labeled as "Figure S5," but it should be "Figure S6." Additionally, the y-axis of the left figure reads "odds of OSA," and the figure legend refers to "Figure S5: Seasonal variation in the odds of OSA (AHI ≥ 15 – left)." This is not accurate and is inconsistent with the manuscript text, which clearly states: "Finally, we reproduced the main analysis by estimating the probability of OSA status (moderate to severe: AHI ≥ 15 ; severe OSA: AHI ≥ 30) across the year."

For consistency and clinical clarity, I recommend revising the legend and y-axis to: "Odds of moderate to severe OSA"

3. The manuscript states: "There may also be implications in terms of safety, since the odds of having moderate to severe OSA varies by $\pm 8\%$ seasonally." It is unclear how this $\pm 8\%$ was derived, and it does not appear to match the seasonal variation shown in the left panel of Figure S6 (previously mislabeled as S5).

4. The authors' approach of comparing exposure–response curves between unadjusted and fully adjusted models is a reasonable strategy within a nonlinear modeling framework, and the visual similarity of these curves may provide some reassurance against major multicollinearity. However, this approach alone is not sufficient to rule out multicollinearity—especially between closely related sleep metrics such as Δ TIB and Δ TST. The example in Supplementary Figure S2 illustrates potential collinearity: the observed association between Δ TIB and AHI largely disappears after adjusting for Δ TST, indicating that the two variables may capture overlapping variance. This weakens the adequacy of relying solely on curve similarity to dismiss multicollinearity concerns in other parts of the analysis.

To strengthen their interpretation, I recommend that the authors include basic correlation matrices or variance inflation factors (VIF)—even from simplified linear models—to better quantify the degree of collinearity. At the very least, a more explicit acknowledgment of this limitation in the methods or discussion would enhance the transparency and robustness of the analysis.

Version 2:

Reviewer comments:

Reviewer #2

(Remarks to the Author)

All of my concerns have been addressed, and I have no further comments.

Dear editor and reviewers,

Thank you for your handling of our manuscript. We are grateful to the reviewers for their helpful comments and suggestions on this manuscript. We hope our manuscript is now acceptable for publication in Communications Medicine.

Regards,

Bastien, on behalf of the authors' team.

Reviewer #1 (Remarks to the Author):

This study presents an extensive exploratory analysis with potentially intriguing associations among seasonal pattern and environmental factors. However, the methodological framework requires refinement to ensure analytical transparency and reproducibility.

Response: Thank you for your comments, especially surrounding the methods, which we believe made the manuscript much better and more reproducible.

Major Concerns:

1. The methodological description of the nonlinear mixed-effects model implementation remains ambiguous. Given that the cited gnm package primarily estimates fixed effects, please provide the complete model specification code in R

Response: Thank you for pointing this out, we made a mistake here and stated “mixed effects” when the case-time-series models methodology that we follow are fixed effects models (we have clarified in the methods). We also now include main model specification R code in supplementary material as suggested.

2. (Line 74) Clarify the geospatial resolution of participant locations (individual addresses vs. aggregated regional data)

Response: Unfortunately, we can not access individual addresses due to privacy and re-identification concerns. We have clarified this point specifically in the methods as follow: “More precise location was not available due to ethical considerations around privacy.”

3. (Line 114) Explicitly state the distribution family (Poisson/Gaussian) and link function for the regression model

Response: We have specified family and link functions in the method as follow: “For continuous outcomes we used the gaussian family with identity link function for the regression models. In some sensitivity analysis with a binary outcome, we used binomial family with logit link function.”

4. (Line 158) Consider constraining the temporal smoothness parameters through: To reduce degrees of freedom for seasonal splines or choose best df using AIC/BIC criteria

Response: This is a good idea. We used AIC with 3 different splines degrees of freedom (4,6 and 8). 8 degrees of freedom seemed to provide the best model fit (see Figure S4), but all models provided similar exposure-response curve. The new sensitivity analysis is described in the methods as follow: “Secondly, we reproduced the main analysis by reducing the degree of freedom for the seasonal splines to 4 and 6.”

And in the results as follow: “The exposure-response curve was also similar when we use more restrictive degrees of freedom for the seasonal spline (Figure S4)”

5. (Line 171) Provide mathematical formulation for Δ AHI calculation

Response: We have included the equation for AHI change in the methods as follows: “Our main outcome of interest was change in AHI (referred to as “AHI change” throughout this manuscript) between a given night and yearly average, expressed as a percentage of the yearly AHI for each participant using the following equation:

$$cAHI_{d,y,p} = 100 * \frac{AHI_{d,y,p} - \overline{AHI}_{y,p}}{\overline{AHI}_{y,p}}$$

Where $cAHI_{d,y,p}$ represents the change in AHI, in %, for a day d , a year y , and participant p and $\overline{AHI}_{y,p}$ represents the averaged AHI for a given year y and participant p .”

We also specifically mention in the Figure 3 legend that “Seasonal effect is summarized using the mean (95%CI) difference between the peak and the trough of the seasonal variation in OSA severity, referred to as ΔAHI .”

6. (Line 149) Correct typographical error ("30-")

Response: Corrected.

7. (Figure 4) Implement dual enhancements: a) Annotate the chromatic scheme with a descriptive legend b) Optimize y-axis scaling through variable-specific ranges rather than uniform limits

Response: We have modified the figure to add the legend. However, we would prefer to keep consistent y-axes across figures as we think these are more suited to compare the effect of the different environmental variables on AHI severity.

Reviewer #2 (Remarks to the Author):

This study examines the seasonal variations in the severity of obstructive sleep apnea (OSA) using large-scale data from Withings Sleep Analyzer users, assessing the impact of environmental factors such as temperature, humidity, and air pollution on the apnea-hypopnea index (AHI). The findings provide valuable insights into seasonal influences on OSA and may have implications for clinical management. However, several critical issues need further clarification and improvement to enhance the study's scientific rigor and applicability.

Response: Thank you for reviewing our manuscript.

1) The study reports a 5-15% seasonal variation in AHI but does not provide actual values, which may lead to misinterpretation. For example, given that the average AHI is 18, this corresponds to a change of 0.9 to 2.9 events/h. Providing these absolute values would help contextualize the findings.

Response: We agree, and we now provide the absolute values in sensitivity analyses. While we agree it may not be too relevant at the individual level, we think our results are still relevant at the population level in terms of safety. Indeed, the odds of moderate to severe OSA is increased by $\pm 8\%$ seasonally. Given the association between OSA and decreased wellbeing, and increased likelihood of motor-vehicle accident(s), these variations may be relevant at the population level. The new added sensitivity analyses are described as follow in the methods: “We also reproduced the main analysis using absolute values of AHI as an outcome within each OSA severity category. Finally, we reproduced the main analysis by estimating the probability of OSA status (moderate to severe: $AHI \geq 15$; severe OSA: $AHI \geq 30$) across the year.”

And the results are described as follow: “The seasonal changes in absolute AHI were consistent in shape between different OSA categories (Figure S5) compared to the main analysis. The absolute

changes were, however, greater in moderate to severe OSA and severe OSA vs. mild OSA (Figure S5). While the changes in absolute AHI at the population may appear small (<2 events/hour), the odds of presenting with moderate to severe OSA (Figure S6a) or severe OSA (Figure S6b) is increased by 10 to 15% in summer and winter compared to autumn and spring.”

We also have modified the discussion to incorporate these points “Therefore, while the observed effect is unlikely to be clinically meaningful at an individual level, at the population level it may lead to misestimation of OSA severity which could have implications for clinical trials. There may also be implications in term of safety, since the odds of having moderate to severe OSA varies by $\pm 8\%$ seasonally.”

2) Percentage changes in AHI may exaggerate seasonal effects in mild OSA while underestimating them in severe OSA, as the relative impact differs depending on baseline severity.

Response: We went back and forth with this issue too. We hope this concern is now fixed with the additional sensitivity analyses above.

3) I suggest using standardized AHI metrics (e.g., Z-score) to ensure comparability across OSA severity groups and provide a more intuitive measure of AHI variability.

Response: This is a good idea. However, using z-score in the context of repeated measurements per participants is not so straightforward. Would one use within participant distribution to estimate the standard deviation or across different severity categories? Within participants may lead to bias estimates, since some participants may be more variable than other [1]. Overall, we think the AHI variability (in %) together with the new addition of absolute AHI and OSA status may be easier to understand.

[1] Lechat, B., et al. (2023). "High night-to-night variability in sleep apnea severity is associated with uncontrolled hypertension." *NPJ Digit Med* 6(1): 57.

2. Selection Bias in the Study Population: T The study is based on Withings Sleep Analyzer users, who are likely to be more health-conscious and of higher socioeconomic status, potentially leading to an underestimation of OSA prevalence and severity in lower-income populations. The limitations section should explicitly acknowledge this selection bias and discuss its implications for the generalizability of the findings. If data are available, I suggest analyzing AHI seasonal variation across different socioeconomic groups to assess potential disparities.

Response: The estimated OSA prevalence from the under-mattress sensor is similar to other previously published estimates, which we mention in the discussion: “Furthermore, OSA prevalence estimates using non-contact multi-night data yield very similar findings to previously published literature^{1,2}.”

We agree with the reviewer regarding the high-socio-economic bias of our sample, which we now discuss as follow: “Most participants also resided in highly developed countries, so they may have also had access to more favourable sleeping environments and heat stress-mitigation strategies such as air conditioning⁴⁶. This high socio-economic bias is common in the sleep research literature⁴⁷. Our results highlight the urgent need for global strategies to collect appropriate sleep and temperature data worldwide⁴⁷.”

Unfortunately, we do not have data on socio-economic status, which we mention in the discussion as follow: “Information on participants socio-demographics was also unavailable.”

3. Collinearity Among Environmental Variables: The study assesses the impact of temperature, humidity, and air pollution on AHI, but these variables are likely highly correlated, which may

introduce collinearity issues in the analysis. I recommend using Principal Component Analysis (PCA) or LASSO regression to reduce collinearity and identify the most influential environmental factors.

Response: We were careful about considering potential collinearity between environmental variables as we approached our *a priori* analysis plan. However, we can see that our considered approach to collinearity is not as well expressed as it could have been in the manuscript. Specifically, to assess collinearity, we ran a minimally adjusted model (with only the environmental exposure of interest) and a fully adjusted model (with other environmental variables) and compared exposure-response curves. As can be seen in Figure 4, there are negligible differences between unadjusted and adjusted models for all environmental variables, suggesting multicollinearity is not an issue. We have made this clearer in the methods: “We compared exposure-response curves between the minimally adjusted models and the fully adjusted models to assess collinearity between variables.”

And in the results: “The associations were similar in unadjusted vs. fully adjusted models suggesting that multi-collinearity between environmental variables was not an issue.”

We have also modified the legend of Figure 4 to make this point clearer: “Figure 4: Associations of different environmental and sleep-specific factors with seasonal variation in the apnoea-hypopnoea-index (AHI) for unadjusted (blue) and fully adjusted models (black). Notes that the associations is similar in unadjusted vs. fully adjusted models suggesting that multi-collinearity between environmental variables is not an issue a) Day of the year (21st of June as reference), b) 24h average temperature, c) density of particulate matter with diameter of less than 2.5µm d) relative humidity e) surface pressure, f) wind speed g) difference between a given night’s total sleep time (TST) with the yearly TST average (Δ TST) in minutes and h) similar to g) but for time in bed (TIB) – see Figure S2 for extra analyses on potential collinearity with Δ TST . All graphs represent estimated marginal means using the 50th percentiles as the reference value (except for a)), and the x-axis limits were set as the 1st percentile and the 99th percentiles.”

PCA or LASSO regression are not appropriate to implement in a fixed-effects paradigm with non-linear associations modelled as spline between environmental variables and the outcome of interest; and would be difficult to interpret.

4. Additionally, interactions between temperature, humidity, and air pollution should be explored.

Response: This is a good suggestion, and we have considered the suitability of adding this to an already substantive manuscript. After careful consideration, and in light of the extensive additions we have already made to this work, we do feel it is beyond the scope of this manuscript, which was aimed at describing the seasonal aspect of OSA severity, rather than an in-depth investigation of environmental factors influencing OSA severity. In addition, the present limitations in geo-localisation and indoor vs. outdoor environment will likely underserve a deeper investigation of these factors. Hence, we feel this would be best described in a separate manuscript.

5. Impact of COVID-19 on AHI Variability: T The data collection period (2020–2023) coincides with the global COVID-19 pandemic, but the study does not discuss how the pandemic may have influenced AHI.

Response: We agree with the reviewer. We have added a new sensitivity analysis to deal with this issue, described as follow in the methods: “To further validate our findings, we conducted several sensitivity and supplementary analyses. Firstly, we reproduced the analysis only in data after September 2022, a period where COVID19 was less likely to confound the observed results.”

The results suggest that COVID19 pandemic was unlikely to be a major confounder in the seasonal aspect of OSA severity and is described as follow in the results: “The exposure-response curve

between the day of the year and the AHI change was similar to the main analysis when data was restricted to before or after September 2022 (Figure S3).”

1) The pandemic increased the risk of OSA through multiple mechanisms. Beyond direct infection effects, pandemic-related lifestyle changes, including weight gain, reduced physical activity, psychological stress, and limited healthcare access, may have exacerbated OSA severity and could confound the observed seasonal effects.

Response: Please see our response above that shows that seasonal variability of OSA is similar in the main analysis vs during covid vs post COVID. It is plausible that the onset of COVID19 may have altered the seasonal pattern in OSA severity/prevalence, however we are unfortunately unable to explore this as we lack data prior to the pandemic.

2) COVID-19 itself exhibits strong seasonal patterns, with peaks often occurring in colder months and coinciding with major holidays when social gatherings increase. These seasonal surges in infection rates may contribute to changes in sleep quality, respiratory health, and indirectly, OSA severity. This factor should be considered in interpreting the study’s seasonal findings.

Response: We agree with the reviewer and in addition to the responses to the points above, we have added the following to the discussion: “Other potential explanations for increased AHI during winter may be increased nasal resistance due to respiratory illnesses such as the flu or COVID-19.”

3) "Long COVID" could have lasting effects on respiratory function, potentially influencing AHI variability independently of seasonal changes.

Response: We agree, and we have added these limitations in our discussion as follow: “Our results remained consistent once data were restricted to post September 2022, suggesting that the COVID-19 pandemic did not seem to influence our main results. However, we did not have access to clinical symptoms and potential COVID-19 infections and/or long COVID, which could have influenced the overall seasonal variability of OSA severity.”

6. The manuscript does not explain why the thresholds of ≥ 4 nights per week and ≥ 28 nights per year were chosen as inclusion criteria.

Response: We have clarified inclusion/exclusion criteria as follows: “Participants were included if they used their devices regularly, defined as ≥ 4 recordings per week and ≥ 28 valid AHI measurements per year (i.e., ≥ 5 hours sleep, see below), similar to previous studies^{2,20,21}.”

Dear editor and reviewer,

Thank you for your handling of our manuscript. We are grateful to reviewer 2 for their additional comments, suggestions and pick-ups in our manuscript. We hope our manuscript is now acceptable for publication in Communications Medicine.

Regards,

Bastien, on behalf of the authors' team.

Reviewer #2 (Remarks to the Author):

I appreciate all the revisions, and the additional sensitivity analyses the authors have provided. However, there are still several issues that should be addressed to improve clarity and consistency:

Response: Thank you for your additional suggestions and for pointing out our mistake with the supplementary figure labels and ordering. Please note that the figures are now labeled in the order in which they appear in the text, so some supplementary figures have been rearranged.

1. In the first Supplementary Figure S5, the y-axis is titled "AHI (events/hour)", which could be misleading. From the visual pattern, the values appear to represent Δ AHI (events/hour), rather than the actual absolute AHI values. The authors should clarify this in both the figure caption and axis labels.

Response: Yes we agree and have modified the figure and label the y-axis as "AHI change, events/hours" to keep it consistent with the other figures. We also have added the following to the caption: "*Absolute AHI change in events/hours was calculated as the difference in AHI between a given night and the yearly average for each participants, and corresponds to $AHI_{d,y,p} - \overline{AHI}_{y,p}$ in Eq (1) of the manuscript.*"

In the right figure of Supplementary Figure S5, it is unclear why the green curve corresponds to the AHI range (-1, 15]. If this is meant to group together participants with no OSA and mild OSA, the lower bound should begin at 0, not -1. Furthermore, the cut-off points at 15 and 30 are standard clinical thresholds for moderate and severe OSA, respectively, yet the figure legend appears to assign 15 and 30 as thresholds for mild and moderate OSA. I suggest revisiting the category labels to align with standard clinical definitions and rerun the analysis if you misclassified them.

Response: Yes we agree, the previous legend was unclear. We have now modified to "Mild OSA", "Moderate to severe OSA", and "Severe OSA", which is in line with the methods.

Additionally, in the stratified analysis by OSA severity, it would be helpful to report the sample size, mean AHI, and range of AHI change for each group. I suggest including a summary table in the supplementary material to improve clarity and transparency. This would also aid in interpreting the following figure (Figure S6).

Response: Yes we agree, we have added a Table S3, which gives the sample size and mean [95%CI] of seasonal change for each group.

2. The second figure in the supplementary material is also labeled as "Figure S5," but it should be "Figure S6."

Response: Thank you for picking up this typo.

Additionally, the y-axis of the left figure reads "odds of OSA," and the figure legend refers to "Figure S5: Seasonal variation in the odds of OSA (AHI \geq 15 – left)." This is not accurate and is inconsistent with the manuscript text, which clearly states: "Finally, we reproduced the main analysis by estimating the probability of OSA status (moderate to severe: AHI \geq 15; severe OSA: AHI \geq 30)

across the year.” For consistency and clinical clarity, I recommend revising the legend and y-axis to: “Odds of moderate to severe OSA”

Response: Yes we agree with your suggestions and have modified the y-label as suggested to “Odds of moderate to severe OSA”.

3. The manuscript states: “There may also be implications in terms of safety, since the odds of having moderate to severe OSA varies by $\pm 8\%$ seasonally.” It is unclear how this $\pm 8\%$ was derived, and it does not appear to match the seasonal variation shown in the left panel of Figure S6 (previously mislabeled as S5).

Response: The $\pm 8\%$ was derived from the Figure S6 by looking at the changes in odds of OSA status between the yearly average and the peak (or trough) of seasonal changes. However, we agree with the reviewer that it is confusing and have rephrased the discussion as follow:

“There may also be implications in term of safety, since the odds of having moderate to severe OSA varies by 15% seasonally, with a peak during summer and winter compared to spring or autumn.”

We also now explicitly mention mean [95%CI] of seasonal changes of OSA status in the results section as follow: “While the changes in absolute AHI at the population may appear small (<2 events/hour), the odds of presenting with moderate to severe OSA (peak to trough differences, see Figure S5) or severe OSA was increased mean [95%CI]; 15.3 [14.7, 16.0]% and 18.7 [17.9, 19.6] % seasonally, respectively.”

4. The authors' approach of comparing exposure–response curves between unadjusted and fully adjusted models is a reasonable strategy within a nonlinear modeling framework, and the visual similarity of these curves may provide some reassurance against major multicollinearity. However, this approach alone is not sufficient to rule out multicollinearity—especially between closely related sleep metrics such as ΔTIB and ΔTST . The example in Supplementary Figure S2 illustrates potential collinearity: the observed association between ΔTIB and AHI largely disappears after adjusting for ΔTST , indicating that the two variables may capture overlapping variance. This weakens the adequacy of relying solely on curve similarity to dismiss multicollinearity concerns in other parts of the analysis.

Response: We discussed the multi-collinearity between ΔTIB and ΔTST in figure legend of Figure S2, which was hidden and probably led to. We now have moved these results to the main manuscript as follow: “Indeed, once adjusting for variation in total sleep time, the association between earlier than normal time-in-bed with AHI change disappear (Figure S7 red vs black curve). This suggests that there was multi-collinearity between variations in sleep timing and variation in sleep duration, similar to a previous study³².”

To strengthen their interpretation, I recommend that the authors include basic correlation matrices or variance inflation factors (VIF)—even from simplified linear models—to better quantify the degree of collinearity. At the very least, a more explicit acknowledgment of this limitation in the methods or discussion would enhance the transparency and robustness of the analysis.

Response: We believe using simpler linear models to be able to calculate VIF is not a good solution here given the highly non-linear associations (J-shaped and U-shaped) between environmental and sleep variables with AHI changes. However, we now provide the correlation matrix between different environmental variables in Figure S6, discussed in the results as follow: “There was minimal correlations between environmental and sleep variables (Figure S6).”

We also have added the following sentence as a limitation in the discussion: “The above limitations, and previous studies showing correlation between air pollution and temperature⁴⁸, or sleep duration

and timing³², may have reduced our ability to determine associations between environmental and sleep-related variables and changes in AHI.”